# Minimum dietary diversity and its associated factors among children aged 6–23 months in Senegal: Evidence from the demographic and health survey, 2023

Enyew Getaneh Mekonen🆔*

Department of Surgical Nursing, School of Nursing, College of Medicine and Health Sciences, University of Gondar, Gondar, Ethiopia

* enyewgetaneh12@gmail.com

## Abstract

### Background

Children between the ages of 6 and 23 months should be provided a variety of foods to ensure that their nutrient needs are satisfied, according to World Health Organization guidelines for feeding children. In Senegal, lower dietary diversity and an inability to meet nutrient requirements raise the risk of anemia and other disorders caused by a lack of funds and resources, which are essential for consuming a nutrient-rich diet. The objective of this study was to assess minimum dietary diversity and its associated factors among children aged 6–23 months in Senegal using the most recent demographic and health survey data (2023).

### Methods

A cross-sectional study was conducted in Senegal using data from the most recent continuous demographic and health survey, 2023. A total of 2,995 children aged 6–23 months were included in the study. Using STATA Version 14 software, the data were extracted, cleaned, and analyzed. A multilevel mixed-effects logistic regression analysis was used to determine the factors associated with meeting minimum dietary diversity. Finally, variables with a *p*-value less than 0.05 were declared statistically significant.

### Results

In the present study, the proportion of children aged 6−23 months who meet minimum dietary diversity in Senegal was 23.61% (95% CI: 22.12, 25.16). Factors like media exposure [AOR = 1.51; 95% CI (1.08, 2.13)], household wealth index [AOR = 1.84; 95% CI (1.24, 2.74)], ANC visits [AOR = 1.50; 95% CI (1.05, 2.13)], current breast-feeding status [AOR = 1.82; 95% CI (1.37, 2.43)], age of the child [AOR = 7.47; 95%

**Data availability statement:** The data from the DHS is publicly available online at https://dhsprogram.com/data/available-datasets.cfm.

**Funding:** The author(s) received no specific funding for this work.

**Competing interests:** The authors have declared that no competing interests exist.

**List of abbreviations:** ANC: Antenatal Care; AOR: Adjusted Odds Ratio; CI: Confidence Interval; DHS: Demographic and Health Survey; ICC: Intra-class Correlation Coefficient; MDD: Minimum Dietary Diversity; MOR: Median Odds Ratio; PCV: Proportional Change in Variance; VIF: Variance Inflation Factor; WHO: World Health Organization.

CI (5.43, 10.3)], number of under-5 children in the household [AOR = 3.97; 95% CI (1.38, 11.4)], and community-level media exposure [AOR = 1.41; 95% CI (1.03, 1.93)] were significantly associated with meeting minimum dietary diversity.

## Conclusions

Nearly one in four children aged 6 to 23 months achieved minimum dietary diversity in Senegal. Children from mothers with media exposure, wealthier households, attending 4 + antenatal care visits, currently breastfeeding, older children, and no under-5 children in the household were associated with higher odds of meeting minimum dietary diversity. Therefore, improving media access, women's empowerment, enhancing antenatal care service utilization, encouraging continued breastfeeding, and giving prior attention to younger children aged 6–11 months and mothers who had under-five children in the household are strongly recommended.

## Background

For children to grow and develop to the best extent possible, the proper meals must be consumed at the right times. When it comes to nutrition, the 1,000 days from conception to the child's second birthday are the most important [1]. Children need to start eating their first foods around six months of age. Throughout the day, young children should be fed regularly, in sufficient amounts, and with a range of food groups in their meals that are high in nutrients. In addition to using clean hands and dishes during meal preparation and serving, caregivers should engage with their children to acknowledge their nutrition requirements [2].

Children between the ages of 6 and 23 months should be provided a variety of foods to ensure that their nutrient needs are satisfied, according to World Health Organization (WHO) guidelines for feeding breastfed and non-breastfed children [3,4]. Variety in food groups is linked to better linear growth in early children [5]. Micronutrient deficiencies may be more likely in a diet low in diversity, which could be detrimental to children's physical and mental development [6]. According to a previous study, stunting was associated with little to no ingestion of nutrient-dense foods such as eggs, dairy products, fruits, and vegetables between the ages of 6 and 23 months [7].

Feeding infants and young children involves more than just making sure they get enough nourishment. Meals are social and cultural gatherings when young children learn about food preferences and dislikes, observe, and mimic behavior that will shape their eating habits for the rest of their lives [8]. A child learns to handle food during meals and makes the connection between food tastes, appearances, and textures. Early child feeding should ideally encourage a child's independence, but it can also be used to control behavioral issues or give a child too much, which can have long-term negative effects on the child's nutrition and health [9]. It's crucial to pay attention to how a child is fed in addition to what they eat [10].

To make it practical and relevant for Member State reporting, the WHO-UNICEF Technical Expert Advisory Group on Nutrition Monitoring (TEAM) suggested in June 2017 revising the minimum dietary diversity (MDD) indicator as defined by WHO (2008). The adjustment involved moving the MDD criterion from four of seven categories to five of eight groups and adding "breast milk" as an eighth dietary group [11]. Therefore, if a child consumes five or more of the following food groups, their diet is considered diverse: breast milk; grains, roots, and tubers; legumes and nuts; dairy products (milk, yogurt, cheese); flesh foods (meat, fish, poultry, liver, or other organs); eggs; vitamin A-rich fruits and vegetables; and other fruits and vegetables [12].

The relationship between dietary diversity and children's health is important, as it is a protective factor against low nutritional status, which can show up as underweight, which is a combination of stunting and wasting in children; wasting, which is an acute form of undernutrition; and stunting, which is chronic and long-term undernutrition [13–15]. A diverse diet is linked to a decrease in undernutrition in children between the ages of six and twenty-three months [16]. It was also observed that children who abstained from eating eggs, dairy products, and other fruits and vegetables had a higher probability of being underweight, severely stunted, and wasted [17].

In Senegal, lower dietary diversity and an inability to meet nutrient requirements raise the risk of anemia and other disorders caused by a lack of funds and resources, which are essential for consuming a nutrient-rich diet [18]. Due to poor cognitive development, hampered academic performance, and low economic production, inappropriate feeding habits pose serious health risks to children that are hard to reverse in later life [19–21]. The prevalence of MDD was 30.8% in Senegal, and the socioeconomic barriers to achieving dietary diversity were identified [22]. The high prevalence of micronutrient malnutrition in Senegal makes it imperative to promptly analyze the nutritional adequacy of young children's meals in order to target the most vulnerable groups with suitable interventions [23]. Studies conducted elsewhere showed that media exposure, household wealth index, age of the child, maternal education, number of families, number of under-five children, ANC visits, and residence were determinants of meeting MDD [22,24–30]. The objective of this study was to assess minimum dietary diversity and its associated factors among children aged 6–23 months in Senegal using the most recent demographic and health survey (DHS) data (2023). While previous studies in West African countries, including Senegal, have assessed MDD among children, the current study expands upon prior research by incorporating a broader range of determinants. By examining maternal factors (such as age and marital status), household characteristics (including the sex of the household head and total children ever born), healthcare access (ANC visits, place of delivery, PNC checkup, and breastfeeding status), and community-level influences (literacy, poverty, and media exposure), this study provides a more comprehensive understanding of MDD in children. Additionally, the use of a larger sample size enhances the robustness and generalizability of the findings, making this study an important contribution to the field.

## Methods and materials

### Data sources, study design, and setting

Cross-sectional data from the continuous DHS in Senegal in 2023 was used for the present study. The DHS-C 2023 is the fifth edition after the continuous surveys conducted in Senegal between 2012–2014, 2015–2016, 2017–2018, and 2019. Other DHS surveys were conducted in Senegal in 1986, 1992–93, 1997, 2005, and 2010. This edition of the survey used a nationally representative sample of 400 clusters and 8,800 households, with an expected number of 16,142 women aged 15–49 surveyed with success. All women aged 15–49 who are members of households or who have spent the night before the day of the survey in selected households are eligible for the survey. As with the previous DHS, the main purpose of the DHS-C 2023 is to gather information on the health of women and their young children, on fertility, on the knowledge and use of contraceptive methods, and on knowledge and attitudes towards sexually transmitted diseases. The results of the survey are representative for Senegal, for the urban environment and rural areas separately, and for each of the fourteen administrative regions. The National Agency for Statistics and Demography (NASD) has a computer

file of Census Districts (DRs) created as a result of the General Population and Housing Census, Agriculture, and Livestock of 2013 (RGPHAE-2013). The datasets are publicly available from the DHS website, www.dhsprogram.com. The surveys are nationally representative of the country and population-based with large sample sizes [31].

## Source and study populations

The youngest child, age 6–23 months, living with the mother in Senegal comprised the source population, whereas women in the selected primary sampling unit (PSU) comprised the study population. Weighted values, which were computed from the Kids Record (KR file) Senegal DHS 2023 datasets, were utilized to restore the representativeness of the sample data. Finally, this study comprised a weighted sample of 2,995 children aged 6–23 months.

## Sampling procedure

The 2023 DHS-C sample is a random, stratified sample drawn at 2 degrees. The primary unit of survey is the DR as defined for the 2013 Census. Each region was separated into parts to form the sampling strata, and the sample was drawn independently from each sampling layer. A total of 28 sample strata were created. Inside each stratum, the DRs were sorted by administrative units below the region, i.e., departments and districts/communes, etc. This operation had introduced implicit stratification at the level of all administrative units below the region with an implicit stratification of the sample allocation proportional to unit size. Prior to the enumeration of households, each large DR having more than 200 households was divided into segments, of which only one was included in the sample. At the second level, in each of the DRs selected in the first, a fixed number of 22 households had been selected with a systematic probability draw from the newly established lists at the time of enumeration. Only the households drawn are interviewed. Replacements of pre-selected households were not allowed immediately, even if they were not for non-responding households, to avoid bias. Every DHS report could be accessed on the Measure DHS website, which included the comprehensive sampling procedure [32].

## Variables of the study

**Dependent variable.** The outcome variable of this study was the minimum dietary diversity status of the child (adequate/inadequate). Children aged 6–23 months who consumed five or more foods and beverages out of the eight defined food groups during the day or night before the survey were considered as meeting MDD (adequate), and inadequate otherwise. The eight food groups are a) breast milk, b) grains, white/pale starchy roots, tubers, and plantains, c) legumes and nuts, d) dairy products (infant formula, milk, yogurt, cheese), e) flesh foods (meat, fish, poultry, and liver/organ meats), f) eggs, g) vitamin A-rich fruits and vegetables, and h) other fruits and vegetables [33].

**Independent variables.** Independent variables from two sources (variables at the individual and community levels) were taken into consideration for this analysis, as DHS data are hierarchical in nature. Variables included at the individual-level were maternal age (15–24, 25–34, 35–49 years), maternal education (no education, primary, secondary & higher), marital status (unmarried, married), sex of the household head (male, female), total children ever born (1–3, 4–6, ≥ 7), media exposure (no, yes), wealth index (poor, middle, rich), ANC visits (none, 1–3 visits, 4 + visits), place of delivery (home, health facility), currently breastfeeding (no, yes), PNC checkup (no, yes), sex of the child (male, female), age of the child (6–11, 12–17, 18–23 months), birth order (first-born, 2–4, 5+), preceding birth interval (< 24 months, ≥ 24 months), and the number of under-5 children in the household (none, 1–2, 3+). Variables included at the community level were place of residence (urban, rural), community literacy (low, high), community-level poverty (low, high), and community media exposure (low, high) These variables were included depending on their availability in the DHS and by reviewing previous literature Fig 1.

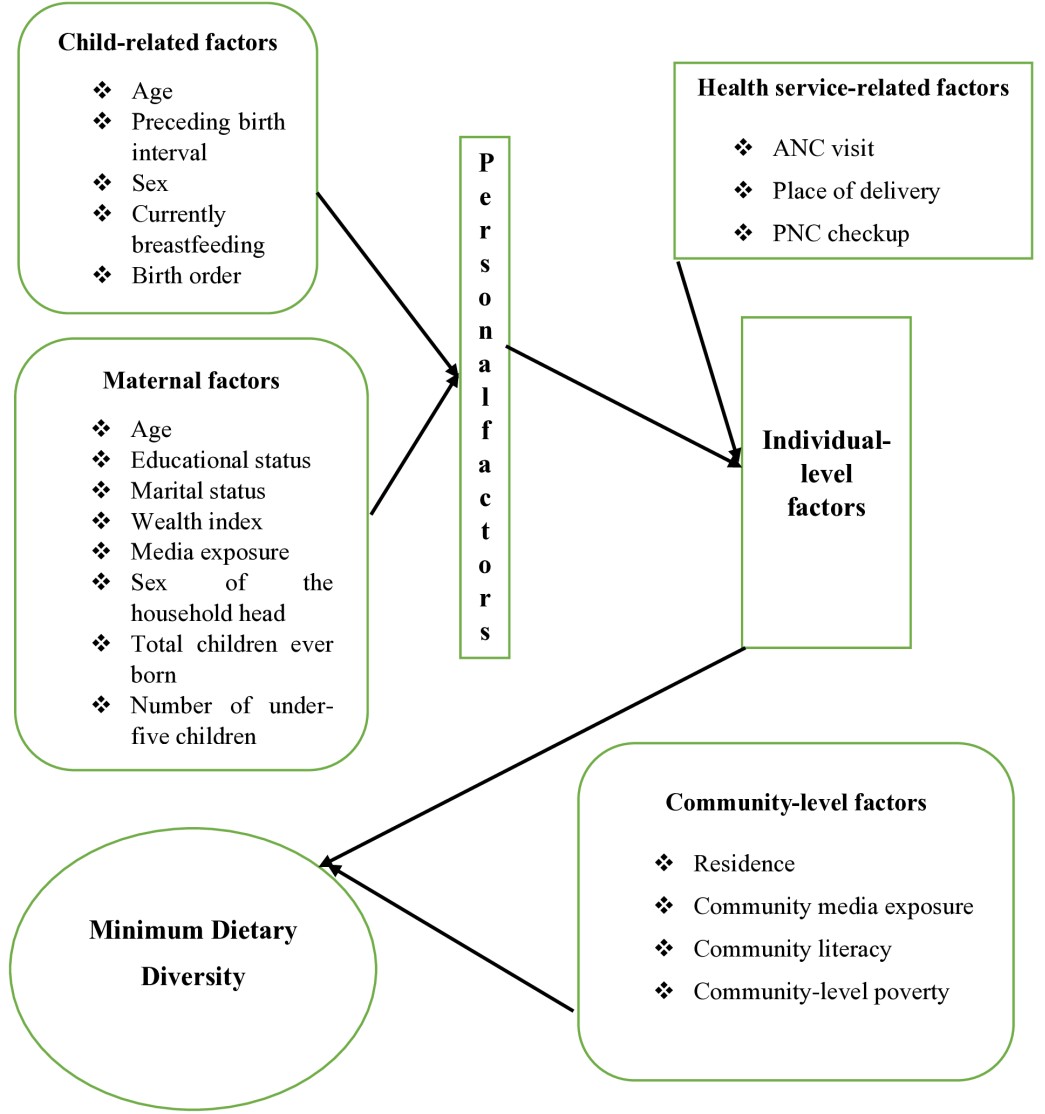

**Fig 1. Conceptual framework for factors associated with minimum dietary diversity among children aged 6 to 23 months in Senegal.**

**Operational description of independent variables.** Media exposure was established by combining three factors: reading newspapers, watching television, and listening to the radio. A woman was deemed to have been exposed to media if she had at least one kind of exposure.

Media exposure at the community level was measured using the percentage of women who had at least been exposed to one of the three media: television, radio, or newspapers. It was coded as "0" for low community-level media exposure (communities where less than 50% of women had at least one media exposure) and "1" for high community-level media exposure (communities where more than 50% of women had at least one media exposure).

The level of poverty in the community was calculated based on the percentage of women in the lowest and poorest quintiles, as obtained from the wealth index data. The poverty levels were classified as "0" for low (those where less than

50% of women belonged to the lowest wealth quintiles) and "1" for high (those where more than 50% of women belonged to the lowest wealth quintiles) communities.

Women's literacy at the community level was also measured using the percentage of women with at least a primary education. It was coded as "0" for low community-level women's education (communities where less than 50% of women had completed primary school) and "1" for high community-level women's education (communities where more than 50% of women completed primary education at the cluster level).

### Data collection tools and quality assurance

Demographics and health surveys use interviewer-administered questionnaire methodologies to gather data through several questionnaire formats. The DHS guideline defined the missing values in the outcome variables in a straightforward manner [33]. However, as complete case analysis is a preferable way to manage missing data in a cross-sectional study, variables with a missing value greater than 5% in explanatory variables were removed from further analysis. To guarantee quality, the data extractions were carried out by researchers with prior experience working with DHS data.

### Model building for multi-level analysis

A multilevel logistic regression model was fitted because the DHS data is hierarchical and children were nested within a cluster, which goes against the independence and equal variance assumptions of ordinary logistic regression models. For the multi-level analysis, four models were fitted. The null model, which solely included the outcome variable, is employed to examine the variation in MDD throughout the cluster. Individual-level and community-level variables are included in the first (Model I) and second (Model II) multilevel models, respectively. The third model (Model III) fitted MDD simultaneously with factors at the individual and community levels. The log-likelihood and deviance test was used to compare the models with the standard logistic regression model, and the best-fitting model was determined by taking the highest log-likelihood and lowest deviance values of each model into account. Multicollinearity was checked using the variance inflation factor (VIF), and a variable is considered to be multicollinear if its VIF score is 10 or more [34]. However, in this investigation, the mean VIF value of the final model was 1.60, and all variables had VIF values less than three. Finally, variables with a p-value of < 0.05 and 95% confidence intervals were declared statistically significant. Proportional Change in Variance (PCV), Intra-class Correlation Coefficient (ICC), and Median Odds Ratio (MOR) were used to evaluate the random effect utilized to measure the variation [35].

### Ethical statement

Permission was granted to download and use the data from http://www.dhs.program.com before conducting the study. Ethical clearance was obtained from the Institutional Review Board of the DHS Program, ICF International. The procedures for DHS public-use data sets were approved by the Institutional Review Board. Identifiers for respondents, households, or sample communities were not allowed in any way, and the names of individuals or household addresses were not included in the data files. The number for each EA in the data file does not have labels to show their names or locations. There were no patients or members of the public involved since this study used a publicly available data set.

## Results

### Individual- and community-level characteristics of study subjects

The mean age of mothers was 28.12±0.13 years, and 42.27% of them fall in the age range of 25–34 years. More than half (54.79%) of mothers had no formal education, and only 5.51% of them were unmarried. More than three-fourths (79.33%) of mothers were from male-headed households, and 61.07% of them had three or fewer children ever born. More than half (57.83%) of mothers had poor household wealth status, and 82.57% of them had media exposure. More

than two-thirds (67.91%) of mothers had four or more ANC visits during their pregnancy, and 81.97% of them currently breastfed their children. The majority (90.52%) of mothers gave birth at a health facility, and 57.77% of them had no postnatal checkup. The mean age of children was 14.18±0.09 months, and 38.09% of them fall in the age range of 12–17 months. More than half (51.59%) of children were male, and 24.54% of them were first-born children. The preceding birth interval for almost all children (99.40%) was ≥24 months, and 55.73% of mothers had three or more under-five children in the household. More than one-third (35.43%) of mothers reside in urban areas, and 50.62% of them were from communities with low levels of literacy. Nearly two-thirds (66.58%) and 55.16% of mothers were from communities with high levels of media exposure and low levels of poverty, respectively [Table 1].

### Prevalence of adequate minimum dietary diversity among children aged 6–23 months

In the present study, the proportion of children aged 6–23 months who meet minimum dietary diversity in Senegal was 23.61% (95% CI: 22.12, 25.16) Fig 2.

The highest proportion of children who meet MDD were found in the Saint-Louis region (45.12%) and the lowest in the Sedhiou region (9.84%) Fig 3.

The proportion of MDD was 89.68%, 31.40%, 73.27%, 79.92%, and 48.23% among children who had media exposure, rich households, attended 4+ANC visits, were currently breastfeeding, and were aged 12–17 months, respectively [Table 2]. From the eight defined groups, breast milk was consumed by nearly 81% of young children, and only 6.61% of them consumed eggs Fig 4.

### Multi-level analysis of factors associated with MDD

**Model comparison and random effect analysis.** Table 3 illustrates that model III in the multilevel analysis outperforms all other multilevel models and the standard logistic regression model (i.e., the model that included all variables but did not include random effect), as it has the highest log likelihood (−1414.7885) and the lowest deviance (2,830) value. According to the ICC value in the null model, cluster differences were responsible for nearly 14.45% of the variation in MDD among children. The MOR between the higher and lower areas among clusters was 2.03, as shown by the MOR in the null model. Furthermore, both individual and community-level variables accounted for around 21% of the heterogeneity in MDD among children [Table 3].

### Fixed-effect analysis

In the final fitted model (Model III) of the multilevel logistic regression analysis, several individual- and community-level factors were found to be significantly associated with MDD. By accounting for hierarchical data structures, this model provides deeper insights into both individual and contextual determinants of MDD. Factors like media exposure, household wealth index, ANC visits, current breastfeeding status, age of the child, number of under-5 children in the household, and community-level media exposure, were significantly associated. Accordingly, the odds of meeting MDD were 1.51 times higher among mothers of children who were exposed to media compared with those who weren't [AOR=1.51; 95% CI (1.08, 2.13)]. Mothers of children with middle and rich wealth status were 1.58 and 1.84 times more likely to meet MDD than those with poor wealth status, respectively [AOR=1.58; 95% CI (1.12, 2.24)] and [AOR=1.84; 95% CI (1.24, 2.74)]. Mothers of children who attended four or more ANC visits were 1.50 times more likely to meet MDD than those who didn't attend ANC visits [AOR=1.50; 95% CI (1.05, 2.13)]. Breastfeeding children were 1.82 times more likely to achieve MDD compared with their counterparts [AOR=1.82; 95% CI (1.37, 2.43)]. Children aged 12–17 months and 18–23 months were 5.21 and 7.47 times more likely to achieve MDD than those aged 6–11 months, respectively [AOR=5.21; 95% CI (3.94, 6.87)] and [AOR=7.47; 95% CI (5.43, 10.3)]. Mothers of children who had no under-five children in the household were 3.97 times more likely to meet MDD than those who had three or more under-five children [AOR=3.97; 95% CI (1.38,

**Table 1. Individual-and community-level characteristics of study subjects in Senegal; DHS 2023.**

| Variables | Category | Frequency (n) | Percentage (%) |
|---|---|---|---|
| Maternal age | 15–24 years | 1,083 | 36.16 |
| | 25–34 years | 1,266 | 42.27 |
| | 35–49 years | 646 | 21.57 |
| Maternal education | No education | 1,641 | 54.79 |
| | Primary | 564 | 18.83 |
| | Secondary & higher | 790 | 26.38 |
| Marital status | Unmarried | 165 | 5.51 |
| | Married | 2,830 | 94.49 |
| Sex of the household head | Male | 2,376 | 79.33 |
| | Female | 619 | 20.67 |
| Total children ever born | 1-3 | 1,829 | 61.07 |
| | 4-6 | 849 | 28.35 |
| | ≥7 | 317 | 10.58 |
| Media exposure | No | 522 | 17.43 |
| | Yes | 2,473 | 82.57 |
| Wealth index | Poor | 1,732 | 57.83 |
| | Middle | 562 | 18.76 |
| | Rich | 701 | 23.41 |
| ANC visits | None | 295 | 9.85 |
| | 1-3 visits | 666 | 22.24 |
| | 4 + visits | 2,034 | 67.91 |
| Place of delivery | Home | 284 | 9.48 |
| | Health facility | 2,711 | 90.52 |
| Currently breastfeeding | No | 540 | 18.03 |
| | Yes | 2,455 | 81.97 |
| PNC checkup | No | 1,677 | 57.77 |
| | Yes | 1,226 | 42.23 |
| Sex of the child | Male | 1,545 | 51.59 |
| | Female | 1,450 | 48.41 |
| Age of the child | 6–11 months | 982 | 32.79 |
| | 12–17 months | 1,141 | 38.09 |
| | 18–23 months | 872 | 29.12 |
| Birth order | First-born | 735 | 24.54 |
| | 2-4 | 1,466 | 48.95 |
| | 5+ | 794 | 26.51 |
| Preceding birth interval | < 24 months | 18 | 0.60 |
| | ≥ 24 months | 2,977 | 99.40 |
| Under-5 children in the household | None | 20 | 0.67 |
| | 1–2 | 1,306 | 43.60 |
| | 3+ | 1,669 | 55.73 |
| Place of residence | Rural | 1,934 | 64.57 |
| | Urban | 1,061 | 35.43 |
| Community literacy | Low | 1,516 | 50.62 |
| | High | 1,479 | 49.38 |
| Community poverty level | low | 1,652 | 55.16 |
| | High | 1,343 | 44.84 |

*(Continued)*

**Table 1.** (Continued)

| Variables | Category | Frequency (n) | Percentage (%) |
|---|---|---|---|
| Community media exposure | Low | 1,001 | 33.42 |
| | High | 1,994 | 66.58 |

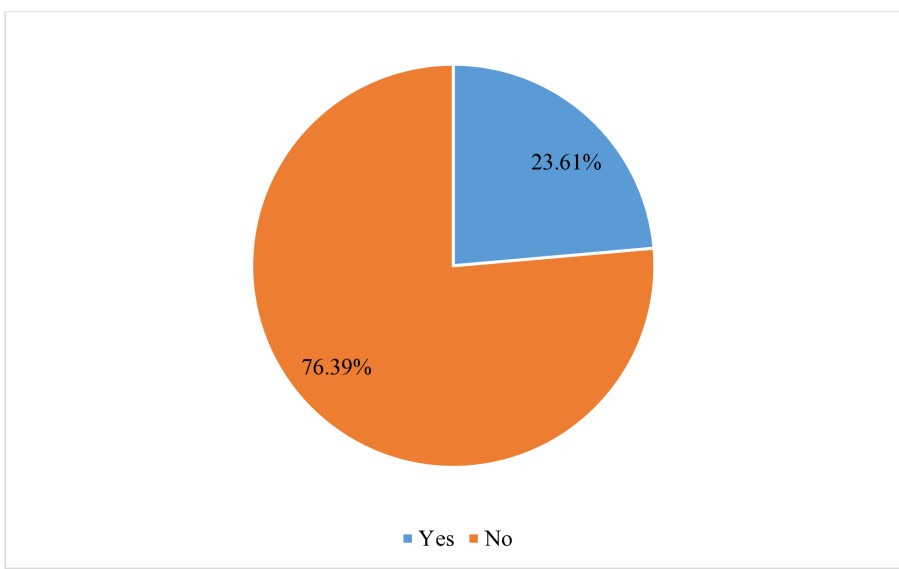

23.61%

76.39%

■ Yes ■ No

**Fig 2. Proportion of children aged 6–23 months who meet minimum dietary diversity in Senegal; DHS 2023 (n = 2,995).**

11.4)]. Mothers of children from communities with high levels of media exposure were 1.41 times more likely to meet MDD compared with their counterparts [AOR = 1.41; 95% CI (1.03, 1.93)] [Table 4].

## Discussion

The long-term burden of stunting among newborns and young children can be significantly decreased by interventions aimed at improving dietary diversity, which determines the nutritional status of children and is a key predictor of stunting [36,37]. In the present study, the proportion of adequate MDD among children aged 6–23 months in Senegal was 23.61% (95% CI: 22.12, 25.16). This finding is consistent with studies conducted in India (22.46%) [38], 32 sub-Saharan African countries (25.1%) [13], and three sub-Saharan African countries (23.2%) [24]. On the other hand, the current finding is higher than studies conducted in Ethiopia (13.4% & 14.9%) [25,39] and Pakistan (20%) [27]. However, this finding is lower than studies conducted in the Gedeo zone, Ethiopia (29.9%) [28], and Bangladesh (38%) [26]. The possible justification for this similarity and discrepancy can be explained by variations in geographical location, study period, seasonal variability, cultural variation, sample size, and other socioeconomic factors.

The study also identified individual and community-level factors significantly associated with MDD. Accordingly, the odds of meeting MDD were higher among mothers of children who were exposed to media compared with those who weren't. Likewise, mothers of children from communities with high levels of media exposure were more likely to meet MDD. This finding is in agreement with studies conducted in Ethiopia [25,39] and three sub-Saharan African countries [24]. The media that women are exposed to, such as reading newspapers, watching TV, and listening to the radio, can encourage them to use maternal healthcare facilities [40,41]. Mass media's informative, instrumental, social control, and community roles in health communication might influence attitudes toward changes in healthy behavior [42]. Children

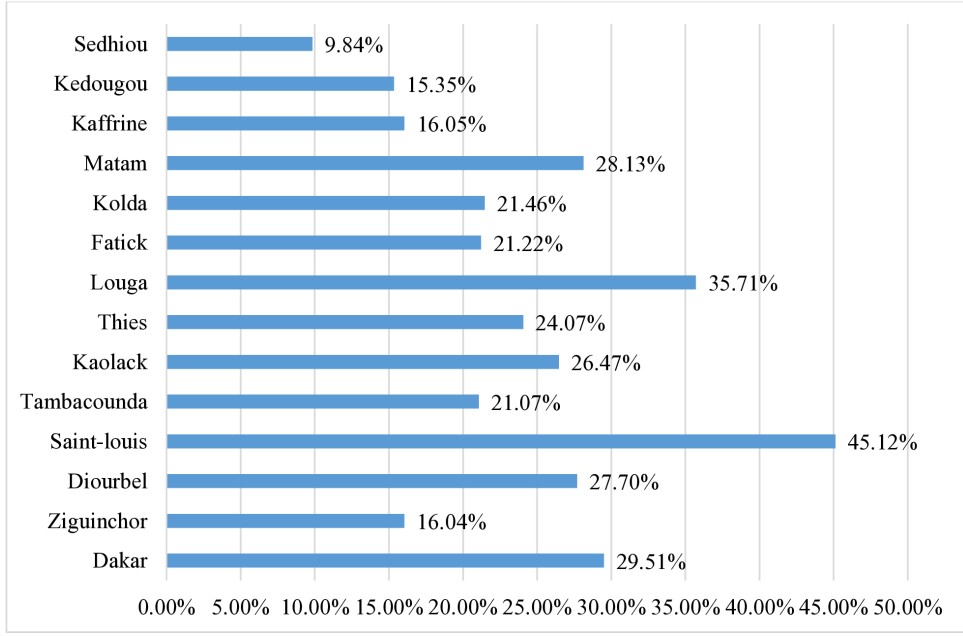

**Fig 3. Regional proportion of children aged 6–23 months who meet minimum dietary diversity in Senegal; DHS 2023 (n = 2,995).**

**Table 2. Patterns of MDD by background characteristics and bivariable analysis.**

| Independent variables | | Meeting MDD | | COR (95%) | *P*-value |
|---|---|---|---|---|---|
| Media exposure | No | No | Yes | 1.00 | |
| | | 19.62% | 10.32% | | |
| | Yes | 80.38% | 89.68% | 2.12 (1.63, 2.76) | <0.001 |
| Wealth index | Poor | 61.41% | 46.25% | 1.00 | |
| | Middle | 17.66% | 22.35% | 1.68 (1.35, 2.09) | <0.001 |
| | Rich | 20.93% | 31.40% | 1.99 (1.63, 2.43) | <0.001 |
| ANC visits | None | 10.36% | 8.20% | 1.00 | |
| | 1-3 | 23.38% | 18.53% | 1.00 (0.71, 1.41) | 0.998 |
| | 4+ | 66.26% | 73.27% | 1.39 (1.03, 1.89) | 0.031 |
| Currently breastfeeding | No | 17.40% | 20.08% | 1.00 | |
| | Yes | 82.60% | 79.92% | 0.84 (0.68, 1.04) | 0.104 |
| Child age | 6-11 months | 39.12% | 12.31% | 1.00 | |
| | 12-17 months | 34.97% | 48.23% | 4.38 (3.40, 5.65) | <0.001 |
| | 18-23 months | 25.91% | 39.46% | 4.84 (3.72, 6.29) | <0.001 |
| Number of under-five children | None | 0.48% | 1.27% | 2.89 (1.19, 7.03) | 0.019 |
| | 1-2 | 42.66% | 46.68% | 1.19 (1.01, 1.42) | 0.040 |
| | 3+ | 56.86% | 52.05% | 1.00 | |
| Community media exposure | Low | 36.54% | 23.34% | 1.00 | |
| | High | 63.46% | 76.66% | 1.89 (1.56, 2.30) | <0.001 |

**COR**: Crude Odds Ratio.

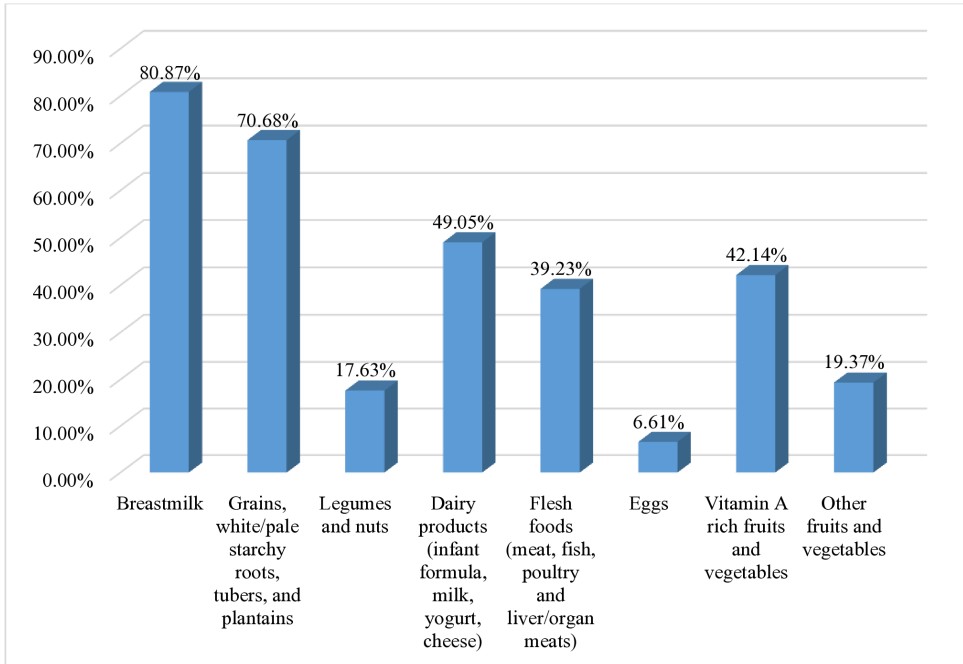

**Fig 4. Proportion of children aged 6–23 months who consumed the eight defined food groups in Senegal; DHS 2023 (n = 2,995).**

**Table 3. Model comparison and random effect analysis for minimum dietary diversity and its associated factors among women in Senegal; DHS 2023 (n = 2,995).**

| Parameter | Null model | Model I | Model II | Model III |
|---|---|---|---|---|
| Variance | 0.555578 | 0.4460063 | 0.4114763 | 0.437621 |
| ICC | 14.45% | 11.94% | 11.12% | 11.74% |
| MOR | 2.03 | 1.88 | 1.84 | 1.87 |
| PCV | Reference | 19.72% | 25.94% | 21.23% |
| Model fitness | | | | |
| LLR | −1608.4565 | −1417.4869 | −1588.1932 | −1414.7885 |
| Deviance | 3,216.913 | 2,834.9738 | 3,176.3864 | 2,829.577 |

ICC: Intra cluster correlation; LLR: Log-likelihood ratio; MOR: Median odds ratio; PCV: Proportional change in variance.

from wealthier households were more likely to achieve MDD. This finding is consistent with studies conducted in Ethiopia [28,39], three sub-Saharan African countries [24], and Bangladesh [26]. This may be explained by the high cost of a variety of foods and the idea that eating them is a luxury. Rather than providing their children with a variety of foods, mothers from low-income backgrounds may also sell their animals and goods in order to get enough money to pay for their children's education, food, and other needs.

Mothers of children who attended four or more ANC visits were more likely to meet MDD than those who didn't. This finding is in line with studies conducted in Indonesia [29] and South Asia [30]. This may be the result of mothers' increased awareness of suitable child feeding practices brought about by the information and counseling they receive from health care providers while using ANC services. This suggests that in order to enhance the eating behaviors of infants and young children, the usage of maternity services should be encouraged. Breastfeeding children were more likely to achieve MDD

**Table 4. Multivariable multilevel logistic regression analysis of factors associated with minimum dietary diversity among children in Senegal; DHS 2023 (n = 2,995).**

| Variables | Category | Model I AOR (95% CI) | Model II AOR (95% CI) | Model III AOR (95% CI) |
|---|---|---|---|---|
| Maternal age | 15–24 years | 1.00 | | 1.00 |
| | 25–34 years | 1.05 (0.81, 1.37) | | 1.05 (0.81, 1.37) |
| | 35–49 years | 0.89 (0.62, 1.30) | | 0.89 (0.62, 1.30) |
| Maternal education | No education | 1.00 | | 1.00 |
| | Primary | 0.92 (0.70, 1.20) | | 0.91 (0.69, 1.19) |
| | Secondary & higher | 0.85 (0.65, 1.11) | | 0.83 (0.63, 1.11) |
| Marital status | Unmarried | 1.00 | | 1.00 |
| | Married | 1.18 (0.74, 1.87) | | 1.19 (0.75, 1.89) |
| Sex of the household head | Male | 1.00 | | 1.00 |
| | Female | 1.09 (0.86, 1.39) | | 1.08 (0.85, 1.38) |
| Total children ever born | 1-3 | 0.76 (0.44, 1.31) | | 0.76 (0.44, 1.31) |
| | 4-6 | 0.75 (0.50, 1.13) | | 0.75 (0.50, 1.13) |
| | ≥7 | 1.00 | | 1.00 |
| Media exposure | No | 1.00 | | 1.00 |
| | Yes | 1.74(1.26, 2.39)* | | 1.51(1.08, 2.13)* |
| Wealth index | Poor | 1.00 | | 1.00 |
| | Middle | 1.67(1.27, 2.21)* | | 1.58(1.12, 2.24)* |
| | Rich | 2.01(1.51, 2.69)* | | 1.84(1.24, 2.74)* |
| ANC visits | None | 1.00 | | 1.00 |
| | 1-3 visits | 1.18 (0.79, 1.76) | | 1.19 (0.80, 1.77) |
| | 4 + visits | 1.50(1.05, 2.13)* | | 1.50(1.05, 2.13)* |
| Place of delivery | Home | 1.00 | | 1.00 |
| | Health facility | 1.16 (0.79, 1.71) | | 1.13 (0.77, 1.66) |
| Currently breastfeeding | No | 1.00 | | 1.00 |
| | Yes | 1.82(1.37, 2.43)* | | 1.82(1.37, 2.43)* |
| PNC checkup | No | 1.00 | | 1.00 |
| | Yes | 1.07 (0.87, 1.31) | | 1.07 (0.87, 1.31) |
| Sex of the child | Male | 1.00 | | 1.00 |
| | Female | 1.05 (0.87, 1.28) | | 1.05 (0.87, 1.28) |
| Age of the child | 6–11 months | 1.00 | | 1.00 |
| | 12–17 months | 5.21(3.95, 6.88)* | | 5.21(3.94, 6.87)* |
| | 18–23 months | 7.46(5.43, 10.3)* | | 7.47(5.43, 10.3)* |
| Birth order | First-born | 0.86 (0.54, 1.38) | | 0.86 (0.54, 1.38) |
| | 2-4 | 0.89 (0.62, 1.29) | | 0.89 (0.62, 1.29) |
| | 5+ | 1.00 | | 1.00 |
| Preceding birth interval | < 24 months | 1.00 | | 1.00 |
| | ≥ 24 months | 0.61 (0.19, 2.03) | | 0.61 (0.18, 2.03) |
| Under-5 children in the household | None | 4.13(1.44, 11.8)* | | 3.97(1.38, 11.4)* |
| | 1–2 | 1.17 (0.96, 1.44) | | 1.17 (0.95, 1.44) |
| | 3+ | 1.00 | | 1.00 |
| Place of residence | Rural | | 1.00 | 1.00 |
| | Urban | | 1.24(0.91, 1.68) | 1.10 (0.79, 1.53) |
| Community literacy | Low | | 1.00 | 1.00 |
| | High | | 1.00(0.78, 1.28) | 1.06 (0.80, 1.40) |

*(Continued)*

**Table 4.** (Continued)

| Variables | Category | Model I AOR (95% CI) | Model II AOR (95% CI) | Model III AOR (95% CI) |
|---|---|---|---|---|
| Community poverty level | low | | 0.77(0.56, 1.06) | 1.09 (0.72, 1.63) |
| | High | | 1.00 | 1.00 |
| Community media exposure | Low | | 1.00 | 1.00 |
| | High | | 1.67(1.26,2.20)* | 1.41(1.03, 1.93)* |

*Statistically significant at P-value<0.05; ANC: Antenatal care; PNC: Postnatal care.

compared with their counterparts. This finding is in agreement with a study conducted in Brazil [43]. A plausible explanation for the correlation between extended breastfeeding and improved feeding practices in later childhood is that families who adhere to the advice to continue breastfeeding for a minimum of two years without providing non-human milk are also likely to prioritize other dietary recommendations for a healthy lifestyle [44].

Likewise, older children were more likely to achieve MDD than younger ones. This finding is consistent with studies conducted in the Gedeo zone, Ethiopia [28], three sub-Saharan African countries [24], and Pakistan [27]. The fact that older children are more accustomed to food than younger children and are more willing to consume a variety of foods with varying tastes and textures could be one explanation for this link [29]. In addition, this outcome may be the result of mothers generally delaying the introduction of cereals and legumes to their very young children (6–11 months) until the child turns one year old [45]. Mothers of children who had no under-five children in the household were more likely to meet MDD than those who had three or more under-five children. This finding is in line with a study conducted in Ethiopia [46]. One explanation could be because mothers with large numbers of under five children are less likely to buy a variety of food categories and are therefore unable to provide for their children's nutritional needs.

### Strengths and limitations of the study

The main strength of the study is utilizing weighted nationally representative data with a large sample size, which makes it representative at the national level. The present study also has sufficient statistical power to generalize the estimates of MDD among children aged 6–23 months during the research period. On the other hand, recall and social desirability bias are likely to arise due to the cross-sectional nature of the data collection method (self-reported). The secondary nature of data was also another limitation of this study, because key variables that affect MDD, like local agriculture, distance to major cities, seasonality, and local climate, were not incorporated in the current analysis.

### Conclusion

Nearly one in four children aged 6 to 23 months achieved minimum dietary diversity in Senegal. Children from mothers with media exposure, wealthier households, who attended 4+ANC visits, who are currently breastfeeding, who are older, and who have no under-5 children in the household were associated with higher odds of meeting MDD. Therefore, improving media access, women's empowerment, enhancing ANC service utilization, encouraging continued breastfeeding, and giving prior attention to younger children aged 6–11 months and mothers who had under-five children in the household are strongly recommended to improve feeding practices of infants and young children.

### Acknowledgments

I am grateful to the DHS program for letting me use the relevant data in this study.

## Author contributions

**Conceptualization:** Enyew Getaneh Mekonen.

**Data curation:** Enyew Getaneh Mekonen.

**Formal analysis:** Enyew Getaneh Mekonen.

**Investigation:** Enyew Getaneh Mekonen.

**Methodology:** Enyew Getaneh Mekonen.

**Software:** Enyew Getaneh Mekonen.

**Supervision:** Enyew Getaneh Mekonen.

**Validation:** Enyew Getaneh Mekonen.

**Visualization:** Enyew Getaneh Mekonen.

**Writing – original draft:** Enyew Getaneh Mekonen.

**Writing – review & editing:** Enyew Getaneh Mekonen.

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
