## [Decision Letter · Decision Letter 0]

7 Mar 2025

PONE-D-24-35929Minimum Dietary Diversity and its Associated Factors among Children Aged 6–23 Months in Senegal: Evidence from the Demographic and Health Survey, 2023PLOS ONE

Dear Dr. Mekonen,

Thank you for submitting your manuscript to PLOS ONE. After careful consideration, we feel that it has merit but does not fully meet PLOS ONE’s publication criteria as it currently stands. Therefore, we invite you to submit a revised version of the manuscript that addresses the points raised during the review process. The second reviewer has suggested major revision in the manuscript and has raised some serious concerns regarding novelty and methodology. These must be addressed adequately, before the manuscript is considered again for PLOS ONE. Please note the Reviewer one has inserted his comments in your manuscript itself. Please be careful to respond to each of them. 

We look forward to receiving your revised manuscript.

Kind regards,

Neetu Choudhary, PhD

Academic Editor

PLOS ONE

Reviewers' comments:

Reviewer's Responses to Questions

**Comments to the Author**

1. Is the manuscript technically sound, and do the data support the conclusions?

Reviewer #1: Yes

Reviewer #2: Partly

2. Has the statistical analysis been performed appropriately and rigorously? 

Reviewer #1: Yes

Reviewer #2: Yes

3. Have the authors made all data underlying the findings in their manuscript fully available?

Reviewer #1: Yes

Reviewer #2: Yes

4. Is the manuscript presented in an intelligible fashion and written in standard English?

Reviewer #1: Yes

Reviewer #2: Yes

5. Review Comments to the Author

Reviewer #1: The article is good. But, should be improved for publication in PLOS One.

My suggestions are in the attached file.

These include suggestions in the background (general research problem, theoretical framework, empirical results, novelty), methods, and results.

All the best.

Reviewer #2: Attached complete review.

6. PLOS authors have the option to publish the peer review history of their article (what does this mean? ). If published, this will include your full peer review and any attached files.

**Do you want your identity to be public for this peer review?** For information about this choice, including consent withdrawal, please see our Privacy Policy .

Reviewer #1: **Yes: ** Omas Bulan Samosir

Reviewer #2: No

---

## [Author Response · Author response to Decision Letter 1]

21 Apr 2025

I have addressed all the comments and attached them as a separate file.

---

## [Decision Letter · Decision Letter 1]

6 May 2025

PONE-D-24-35929R1Minimum Dietary Diversity and its Associated Factors among Children Aged 6–23 Months in Senegal: Evidence from the Demographic and Health Survey, 2023PLOS ONE

Dear Dr. Mekonen,

Thank you for submitting your manuscript to PLOS ONE. After careful consideration and based on reviewers' comments, we feel that certain minor issues must be addressed before the paper may be considered further for publication. Therefore, we invite you to submit a revised version of the manuscript that addresses the points raised during the review process.

We look forward to receiving your revised manuscript.

Kind regards,

Neetu Choudhary, PhD

Academic Editor

PLOS ONE

Journal Requirements:

Reviewers' comments:

Reviewer's Responses to Questions

**Comments to the Author**

1. If the authors have adequately addressed your comments raised in a previous round of review and you feel that this manuscript is now acceptable for publication, you may indicate that here to bypass the “Comments to the Author” section, enter your conflict of interest statement in the “Confidential to Editor” section, and submit your "Accept" recommendation.

Reviewer #1: All comments have been addressed

Reviewer #2: All comments have been addressed

2. Is the manuscript technically sound, and do the data support the conclusions?

Reviewer #1: Yes

Reviewer #2: Yes

3. Has the statistical analysis been performed appropriately and rigorously? 

Reviewer #1: Yes

Reviewer #2: Yes

4. Have the authors made all data underlying the findings in their manuscript fully available?

Reviewer #1: Yes

Reviewer #2: Yes

5. Is the manuscript presented in an intelligible fashion and written in standard English?

Reviewer #1: Yes

Reviewer #2: Yes

6. Review Comments to the Author

Reviewer #1: There are still comments to address.

Should explain the significance of the results of multilevel analysis.

They are in the attached file.

Thank you.

Reviewer #2: (No Response)

7. PLOS authors have the option to publish the peer review history of their article (what does this mean? ). If published, this will include your full peer review and any attached files.

**Do you want your identity to be public for this peer review?** For information about this choice, including consent withdrawal, please see our Privacy Policy .

Reviewer #1: **Yes: ** Omas Bulan Samosir

Reviewer #2: No

---

## [Author Response · Author response to Decision Letter 2]

16 Jun 2025

Use italic for p in p-value.

Thank you very much, and I have corrected it.

Still needs to address the general research problem i.e. that in Senegal the negative impact of inadequate minimum dietary diversity, such as lower education achievement, is a problem, usually addressed in the national development plan.

I have addressed it.

Still needs to address the novelty of this study.

I have addressed it.

Is there any statistical problem using maternal education and this community problem in the same model?

Yes, because cluster number is incorporated to generate community-level variables.

Should have the results of bivariate analysis between all independent variables and dependent variables that show a bivariate relationship, such as the higher the maternal education, the higher the percentage of children who met MDD.

Thank you very much, and I have included the bivariate analysis.

Any explanation about the multilevel analysis results here regarding the outcome dependence within the same communities?

Thank you very much, and I have included an explanation.

Why mothers? How about the children?

Corrected.

Thank you very much for your time and effort.

---

## [Decision Letter · Decision Letter 2]

28 Aug 2025

Minimum Dietary Diversity and its Associated Factors among Children Aged 6–23 Months in Senegal: Evidence from the Demographic and Health Survey, 2023

PONE-D-24-35929R2

Dear Dr. Enyew Getaneh Mekonen,

We’re pleased to inform you that your manuscript has been judged scientifically suitable for publication and will be formally accepted for publication once it meets all outstanding technical requirements.

Kind regards,

Abu Sayeed, MSc

Academic Editor

PLOS ONE

Additional Editor Comments (optional):

Reviewers' comments:

Reviewer's Responses to Questions

**Comments to the Author**

1. If the authors have adequately addressed your comments raised in a previous round of review and you feel that this manuscript is now acceptable for publication, you may indicate that here to bypass the “Comments to the Author” section, enter your conflict of interest statement in the “Confidential to Editor” section, and submit your "Accept" recommendation.

Reviewer #1: All comments have been addressed

Reviewer #2: All comments have been addressed

2. Is the manuscript technically sound, and do the data support the conclusions?

Reviewer #1: Yes

Reviewer #2: Yes

3. Has the statistical analysis been performed appropriately and rigorously? 

Reviewer #1: Yes

Reviewer #2: Yes

4. Have the authors made all data underlying the findings in their manuscript fully available?

Reviewer #1: Yes

Reviewer #2: Yes

5. Is the manuscript presented in an intelligible fashion and written in standard English?

Reviewer #1: Yes

Reviewer #2: Yes

6. Review Comments to the Author

Reviewer #1: Please revise Table 2 and its analysis in the text. Use 100% row total to be meaningful so that one can conclude that the percentage of MDD was higher among mothers who had media exposure and so on.

Do for each category of all independent variables.

Reviewer #2: (No Response)

7. PLOS authors have the option to publish the peer review history of their article (what does this mean? ). If published, this will include your full peer review and any attached files.

**Do you want your identity to be public for this peer review?** For information about this choice, including consent withdrawal, please see our Privacy Policy .

Reviewer #1: No

Reviewer #2: No

---

## [Editor Report · Acceptance letter]

PONE-D-24-35929R2

PLOS ONE

Dear Dr. Mekonen,

I'm pleased to inform you that your manuscript has been deemed suitable for publication in PLOS ONE. Congratulations! Your manuscript is now being handed over to our production team.

Kind regards,

on behalf of

Mr. Abu Sayeed

Academic Editor

PLOS ONE